# Effects of *Sapindus mukorossi* Seed Oil on Bone Healing Efficiency: An Animal Study

**DOI:** 10.3390/ijms25126749

**Published:** 2024-06-19

**Authors:** Po-Jan Kuo, Yu-Hsiang Lin, Yu-Xuan Huang, Sheng-Yang Lee, Haw-Ming Huang

**Affiliations:** 1Department of Periodontology, School of Dentistry, Tri-Service General Hospital and National Defense Medical Center, Taipei 11490, Taiwan; kuopojan@gmail.com; 2Department of Dentistry, Wan Fang Hospital, Taipei Medical University, Taipei 11696, Taiwan; d513111002@tmu.edu.tw (Y.-H.L.); seanlee@tmu.edu.tw (S.-Y.L.); 3Graduate Institute of Injury Prevention and Control, College of Public Health, Taipei Medical University, Taipei 11031, Taiwan; 4School of Dentistry, College of Oral Medicine, Taipei Medical University, Taipei 11031, Taiwan; tges89139@yahoo.com.tw

**Keywords:** tissue engineering, seed oil, fatty acid, stem cell, osteogenesis, animal study

## Abstract

Natural products have attracted great interest in the development of tissue engineering. Recent studies have demonstrated that unsaturated fatty acids found in natural plant seed oil may exhibit positive osteogenic effects; however, few in vivo studies have focused on the use of plant seed oil for bone regeneration. The aim of this study is to investigate the effects of seed oil found in *Sapindus mukorossi* (*S. mukorossi*) on the osteogenic differentiation of mesenchymal stem cells and bone growth in artificial bone defects in vivo. In this study, Wharton-jelly-derived mesenchymal stem cells (WJMSCs) were co-cultured with *S. mukorossi* seed oil. Cellular osteogenic capacity was assessed using Alizarin Red S staining. Real-time PCR was carried out to evaluate ALP and OCN gene expression. The potential of *S. mukorossi* seed oil to enhance bone growth was assessed using an animal model. Four 6 mm circular defects were prepared at the parietal bone of New Zealand white rabbits. The defects were filled with hydrogel and hydrogel-*S. mukorossi* seed oil, respectively. Quantitative analysis of micro-computed tomography (Micro-CT) and histological images was conducted to compare differences in osteogenesis between oil-treated and untreated samples. Although our results showed no significant differences in viability between WJMSCs treated with and without *S. mukorossi* seed oil, under osteogenic conditions, *S. mukorossi* seed oil facilitated an increase in mineralized nodule secretion and upregulated the expression of ALP and OCN genes in the cells (*p* < 0.05). In the animal study, both micro-CT and histological evaluations revealed that new bone formation in artificial bone defects treated with *S. mukorossi* seed oil were nearly doubled compared to control defects (*p* < 0.05) after 4 weeks of healing. Based on these findings, it is reasonable to suggest that *S. mukorossi* seed oil holds promise as a potential candidate for enhancing bone healing efficiency in bone tissue engineering.

## 1. Introduction

Tissue engineering applies both biological and engineering principles to create materials aimed at repairing, maintaining, or enhancing the function of human tissue [1]. Treating extensive bone defects resulting from trauma or tumor resection remains an increasingly daunting clinical challenge today [2]. In recent years, autologous and allogeneic bone transplantation has been the primary method of clinical treatment; however, sourcing limitations for bone grafts, along with complications and potential drawbacks such as immunological rejection, have restricted their application. Recently, an increasing number of biomaterials have been utilized as bone substitutes for repairing bone defects [3]. Among these materials, natural free fatty acids (FFAs) present in plants are considered potential agents for tissue regeneration [4]. Fatty acids are categorized as saturated fatty acids (SFAs) and unsaturated fatty acids (UFAs) based on the existence of double bonds in their chemical structure. UFAs are further classified into two major categories: monounsaturated fatty acids (MUFAs), which contain one double bond and polyunsaturated fatty acids (PUFAs), which contain multiple double bonds. UFAs are also categorized based on the position of the double bond in the carbon chain, including omega-3 (ω-3) at the third carbon position, omega-6 (ω-6) at the sixth carbon position, and omega-9 (ω-9) at the ninth carbon position.

In bone tissue engineering, SFAs have been reported to stimulate osteoclastogenesis, which is detrimental to bone health [5]. UFAs have varied effects due to their differing chemical structures. Studies have reported that a diet rich in PUFAs has a positive effect on bone formation by reducing chronic inflammation and bone loss [6]. However, recent research has found that the effects of ω-3 and ω-6 UFA on bone health are different. Some studies suggest that ω-3 UFA, such as docosahexaenoic acid (DHA) and eicosapentaenoic acid (EPA), significantly increase bone regeneration and improve microstructure and structural strength [7,8,9,10]. Additionally, animal studies have shown that ω-3 UFA not only has anti-inflammatory effects [11] but also reduced osteoclasts by 60% and bone resorption by 80% [12]. Similar results were found with bone marrow stem cells [13]. Martyniak et al. (2021) suggested that consuming ω-3 PUFAs have a positive effect on maintaining bone health in elderly individuals [6], and a review concluded that alpha-linolenic acid (ALA) has the greatest positive impact on bone health among ω-3 UFAs [14]. However, the opposite effects are found with ω-6 PUFAs, with studies showing that ω-6 PUFAs promote inflammation [11] and activate osteoclasts [15], thereby increasing the risk of fractures [6].

ω-9 FA is another type of unsaturated fatty acid that significantly influences bone regeneration. Deshimaru (2005) conducted in vitro cell experiments testing various FAs and found that only oleic acid, an ω-9 MUFA, has the ability to promote osteoblast differentiation [16]. Omer’s team first compared ω-6 and ω-9 FAs on bone regeneration in animal experiments and found that animals fed with ω-9 MUFAs exhibited significant improvements in maintaining bone strength and viscoelastic properties [17]. The authors later suggested that besides promoting bone regeneration, ω-9 MUFAs also have a significant inhibiting effect toward bone loss [18].

In recent years, plant-derived essential oils have garnered significant scientific interest due to their antioxidant, anti-inflammatory, and antimicrobial properties. Numerous studies aim to explore the direct application of essential oils on bone tissue, their incorporation as bioactive compounds in bone scaffolds, or their use as coatings for bone implants [4]. Fatty acids can be found in oil derived from soapnut trees, a cash-crop native to Asian tropical and subtropical climates from Japan to India [19]. The genus Sapindus comprises over 2000 wild species [19], with *Sapindus mukorossi* (*S. mukorossi*) containing approximately 23% oil in the seed kernel (Figure 1a). Chemical composition analysis revealed that *S. mukorossi* seed oil contains a significant amount of fatty acids, with up to 86% being unsaturated fatty acids [20,21]. Cell and animal studies have separately confirmed the oil’s antibacterial and anti-inflammatory effects, both in aiding skin repair [8] and stabilizing oral microbiota [22]. Interestingly, a recent study has also shown that *S. mukorossi* seed oil promotes osteogenic/odontogenic differentiation and matrix vesicle secretion of dental pulp stem cells [23], though the exact mechanism remains unknown. As *S. mukorossi* seed oil has been reported to show potential in bone tissue engineering applications, this study conducted an animal experiment to explore the bone healing ability of this natural plant oil product.

## 2. Results

In this study, the *S. mukorossi* species was confirmed via NMR spectroscopy. The 1H NMR spectrum of *S. mukorossi* seed oil is shown in Figure 1b. As described above, the major NMR characteristic of *S. mukorossi* seed oil can be identified by a chemical shift of δ 0–3 ppm [20]. The chemical shift at δ 0.8–1.0 ppm is due to the terminal methyl protons (C-CH3). Multi-signals between 1.2 and 1.4 ppm are due to the protons along the backbone of the long fatty acid chain. Peaks at δ 1.5–1.7 and 1.9–2.1 ppm are the structure of (-CH2C-CO2) and (-CH2-C=C-), respectively. Multi-peaks at δ 2.2–2.4 ppm are partly a result of methylene protons close to the ester group (-CH2-CO2). The signal close to δ 2.8 ppm indicates the presence of bis-allylic protons (C=C-CH2C=C-) of the polyunsaturated fatty acid chain. The signal between δ 5.3 and 5.6 ppm is contributed to by cyanolipid contained in the oil [24]. Multi-peaks at δ 5.2–5.5 ppm represent olefinic protons (-CH=CH-) in unsaturated fatty acid chains.

FA composition of the *S. mukorossi* seed oil was determined by gas chromatography-mass spectrometry (GC-MS) analysis. Figure 1c and Table 1 show that *S. mukorossi* seeds contain abundant UFAs (89.54%). Among these UFAs, PUFAs and MUFAs account for 10.7% and 89.3%, respectively. The primary MUFAs in *S. mukorossi* seed oil are oleic acid (C18:1) and eicosenic acid (C20:1).

Figure 2 shows the results of cell viability analysis conducted on Wharton-jelly-derived mesenchymal stem cells (WJMSCs) cultured with *S. mukorossi* seed oil. Cell viability steadily increased over the initial five days and reached a plateau after three days. Throughout the entire experimental duration, no significant difference in cell viability was observed between the control group and MSC cells co-cultured with seed oil. The addition of *S. mukorossi* seed oil in the medium does not appear to exert a significant effect on the viability of WJMSC cells.

As depicted in Figure 3, WJMSCs cultured with standard medium exhibited minimal deposition of mineralized nodules (Figure 3a). In contrast, stem cells cultured in osteogenic medium exhibited prominent, dark-red staining, signifying the presence of calcium ion deposits (Figure 3b). Furthermore, in the case of cells co-cultured with *S. mukorossi* seed oil and osteogenic medium, robust positive staining was evident across the entirety of the culture plate (Figure 3c). Figure 3d illustrates a quantitative assessment of the Alizarin Red S staining. The calcium content of standard medium, osteogenic medium, and osteogenic medium with seed oil are recorded as 0.05 ± 0.003, 0.08 ± 0.007, and 0.116 ± 0.028, respectively. A significant increase in calcium ion deposits was found when *S. mukorossi* seed oil was added to osteogenic medium (*p* < 0.05). The significant increase in calcium ion deposits due to the supplementation shows the capacity of *S. mukorossi* seed oil to promote osteogenic differentiation in WJMSCs.

The addition of *S. mukorossi* seed oil for 48 h significantly increased osteogenic gene expression (ALP and OCN) in WJMSCs (Figure 4). The detected cellular expressions when cultured in osteogenic medium with seed oil were 2.73 ± 0.255 and 4.94 ± 0.073 for the ALP (Figure 4a) and OCN genes (Figure 4b), respectively. These values were significantly higher than that for cells cultured in pure osteogenic or normal medium (*p* < 0.05).

Figure 5 presents 3D images obtained from CT analysis of artificial bone defects after 4 weeks. In comparison to control samples (Figure 5a), both hydrogel (HG) (Figure 5b) and hydrogel with *S. mukorossi* seed oil (HG-SM oil) (Figure 5c) show enhanced bone growth. In the HG group, however, complete coverage of the central region by new bone was not achieved in the 4-week healing period (Figure 5b). Nevertheless, the entire defect area was almost covered by newly formed bone in samples treated with HG combined with *S. mukorossi* seed oil (Figure 5c). Quantitative analysis (Figure 6a–c) reveals that new bone formation in defects treated with HG and HG-SM oil complexes was 3.79 (*p* < 0.05) and 6.72 (*p* < 0.01) times higher, respectively, than in the control group at 4 weeks (Figure 6d). Comparable outcomes were also observed when trabecular thickness was considered (Figure 6e). No disparity in trabecular number was noted with the use of HG alone (Figure 6f). Nevertheless, trabecular separation exhibited significant decrease (Figure 6g) when defects were filled with bone grafts containing *S. mukorossi* seed oil.

After 4 weeks of healing, untreated artificial bone defects had filled with connective soft tissue (Figure 7a,b). A comparable outcome was observed in histological experiments. When artificial bone defects were treated with HG alone, newly formed bone was observed close to the wound surface (Figure 7c,d). When defects were treated with hydrogel mixed with *S. mukorossi* seed oil, improved bone healing could be seen. In this situation, more new bone was seen in deep tissue (Figure 7e,f). Compared to the control group (12.77 ± 5.66%), a significant increase (*p* < 0.05) in newly formed bone was found when the artificial defect was filled with HG (32.47 ± 11.09%) and HG-SM oil (52.26 ± 8.17%) (Figure 8a). Defects treated with hydrogel combined with *S. mukorossi* seed oil had 1.6 times more newly formed bone compared to defects treated with hydrogel alone (*p* < 0.05). The percentage of connective tissue was 51.88 ± 8.84% for the HG group and 24.71 ± 6.27% for the HG-SM oil group, significantly lower (*p* < 0.05) than in untreated defect samples (62.63 ± 4.31%) (Figure 8b). No statistical difference was observed throughout the 4-week healing period in the residual bone substitute among the three groups (Figure 8c). These findings further support the conclusion that adding *S. mukorossi* seed oil to bone grafts results in a more effective bone growth process.

## 3. Discussion

Studies have shown that plant extracts have the potential to promote MSC proliferation and differentiation [25,26,27]. The addition of *S. mukorossi* seed oil has also been reported to increase the degree of differentiation in osteogenesis and mineralized nodule deposition of dental pulp stem cells (DPSCs), although no effect on their proliferation has been reported [23]—a phenomenon that was confirmed in the current study (Figure 2). Indeed, several studies have indicated that it is not possible to simultaneously enhance both proliferation and differentiation of osteoprogenitor cells [28,29]. In addition, different plant extracts have varying chemical compositions, which in turn lead to different cellular responses. For example, SFAs and ω-6 UFAs present in vegetable oils could potentially have negative effects on bone regeneration [6,11,15], and another previous study also found that adding palmitic acid (C16:0), a saturated fatty acid, to stem cell culture medium leads to cell apoptosis [30]. As in previous work [8,31], as well as in the present study, *S. mukorossi* seed oil contains only 7.52% ω-6 UFA (primarily linoleic acid) (Table 1) and 3.25% of palmitic acid, which avoid the possible side effects of using *S. mukorossi* seed oil as a material for bone regeneration.

In this study, we found that *S. mukorossi* seed oil significantly increased mineralized nodules of WJMSCs (Figure 3). As mentioned above, ω-3 UFAs significantly increase bone regeneration [7,9,10,32] and can promote bone regeneration by enhancing calcium ion transmission [33] and strengthening at the Akt signaling pathway for cell membranes [34]. However, ω-6 UFAs can activate osteoclasts [15], and several studies have indicated that the ratio of ω-3 to ω-6 UFAs within cell membranes can be an indicator of cellular function [11], particularly in bone regeneration pathways induced by cell membranes [35]. In Table 1, we show that only 2.04% ω-3 UFA (mainly linolenic acid) is contained in *S. mukorossi* seed oil. That is, the addition of *S. mukorossi* seed oil significantly improves the osteogenic differentiation of WJMSCs (Figure 3 and Figure 4), which is not caused by ω-3 UFA.

Rather, we found an abundance of ω-9 MUFAs (79.98%) in *S. mukorossi* seed oil, consistent with previous studies [8,20,33]. The ω-9 MUFAs in *S. mukorossi* seed oil are primarily oleic acid (C18:1, 55.20%) and cis-11-eicosenoic acid (C20:1, 24.1%). Oleic acid is one of the main components of cell membranes [36], the addition of which to cultures of human adipose-derived stem cells, along with osteogenic stimulation, has been shown to result in a significant increase in calcium nodules compared to control [37]. Similar findings were observed in this study (Figure 3). Symmank et al. (2020) found that adding oleic acid to stem cell culture medium can promote differentiation into bone cells [30]. In addition, analysis of differentiation-related genes revealed a significant increase in early bone marker expression and ALP activity. These findings are also consistent with the results shown in Figure 4, where the addition of SM seed oil to WJMSC cultures led to an increase in osteogenic gene expression.

One limitation of using *S. mukorossi* oil for bone healing is its high fluidity, which can cause it to flow out of the defect and reduce its bone growth efficiency. Therefore, manufacturing a hydrogel scaffold to release *S. mukorossi* oil is essential for its effective clinical application. Our in vivo experiments suggest that bone defects treated with filling hydrogel containing *S. mukorossi* seed oil exhibit better reparative processes than compared to untreated analogs or ones with hydrogel only. When viewed in combination with CT data (Figure 5 and Figure 6) and histological results (Figure 7 and Figure 8), a reasonable conclusion can be made that *S. mukorossi* seed oil provides an osteoregenerative effect on bone repair. This finding is consistent with a previous report that used bone graft complexes with poly-ε-caprolactone and oleic acid implanted into bone defects, resulting in more new bone formation and higher bone trabecular density compared to the control [37]. Tatara et al. (2019) conducted animal experiments to investigate the relationship between various saturated and unsaturated FAs and bone density and indicated that only eicosenoic acid and oleic acid showed a positive correlation with bone quality [38]. Specifically, oleic acid was highly correlated with increased bone strength. Since eicosenoic and oleic acids are primary components of *S. mukorossi* seed oil, amounting to approximately 80% of the oil (Table 1), we can reasonably attribute the bone regeneration effect of *S. mukorossi* seed oil to the abundance of these ω-9 MUFAs.

The structure and physicochemical properties of the cell membrane are related to the reception of external signals for carrying out specific functions [39]. Phospholipids in the cell membrane form a bilayer membrane structure with the hydrophobic tails of fatty acids. These fatty acids may consist of SFAs or UFAs. The composition of lipid components in the cell membrane can cause its physical properties to behave in a manner between a fluid and a gel-like phase [40]. Therefore, in biophysics, we can define the physical state of the cell membrane by its membrane fluidity. It has been reported that the composition and characteristics of the stem cell membrane can reflect their differentiation status [41], because normal cell membrane function requires a relative balance between saturated, monounsaturated, and polyunsaturated acyl chains [36]. If fatty acids at the hydrophilic end of phospholipids are saturated, their molecular structure is linear. Consequently, they pack together densely in a given space, resulting in higher order. Conversely, curved unsaturated fatty acid molecules cannot pack tightly together, increasing cell membrane fluidity [40]. It has been reported that the increased rigidity of cell membranes caused by SFAs can have adverse effects on cell survival. In contrast, UFAs such as oleic acid and linoleic acid have less effect on cell membrane fluidity and exhibit cell membrane stabilizing properties [42]. There is evidence to suggest that during the induction of pluripotent stem cell differentiation, cell membrane fluidity undergoes significant changes. For bone tissue engineering, previous studies have also shown that altering cell membrane fluidity due to mechanical deformation of the membrane surface can enhance the osteogenic differentiation ability of dental pulp stem cells [43,44,45,46]. The calvarial defect model used in this study has the advantage of lacking mechanical stress. Therefore, the results of this study can be replicated in bone defects occurring under low mechanical stress conditions. Accordingly, this model can serve as a complementary treatment method for bone defects in situations where mechanical stress is insufficient. On the contrary, this model also has limitations. Given the fact that calvaria bone in rabbits is rather thin, histomorphometric measurements are difficult to perform to demonstrate the growth situation of deep tissue. This issue should be considered in further experiments.

Interestingly, a previous study has found that the addition of oleic acid to cell phospholipids can indeed modulate membrane structure, thereby altering the membrane fluidity within the cell membrane [36]. Additionally, findings of Kurniawan et al. (2017) also indicate that oleic acid can react with saturated lipid molecules on the membrane and insert into the cell membrane structure [47]. This results in a decrease in lipid saturation on the cell membrane, consequently reducing membrane rigidity and increasing membrane fluidity. These results may be the possible reason why *S. mukorossi* seed oil, which is rich in oleic acid, can promote WJMSC differentiation (Figure 3 and Figure 4) and enhance bone growth as observed in the in vivo experiments (Figure 5, Figure 6, Figure 7 and Figure 8).

Although the limited results present in this study demonstrate the possible beneficial effect of *S. mukorossi* seed oil on bone regeneration, further research is necessary to reveal the possible mechanisms, particularly the biophysical properties related to calcium ion transport and enhanced cell membrane fluidity. In addition, the expression profile of osteogenic genes is also important for clarifying the mechanism of this study. These will become the aspects of further, large-scale research.

## 4. Materials and Methods

### 4.1. Preparation of S. mukorossi Seed Oil

The *S. mukorossi* seeds used in this study (Figure 1) were purchased from He He Co., Ltd. (Taipei, Taiwan). Prior to extraction, the seeds were cleaned using running tap water followed by rinsing with sterile distilled water. The seeds were then dried in an oven at 40 °C for a duration of 72 h. Next, the seeds were ground using a grinder, and kernels were separated from their tough shells. Consistent with a prior study [48], the oil was extracted by cold pressing followed by filtration using a 0.45 μm pore size filter.

The plant species was confirmed by identifying the chemical structure through 1H nuclear magnetic resonance (NMR) spectrums obtained from a 500 MHz NMR spectrometer (DRX500 Avance, Bruker BioSpin GmbH, Rheinstetten, Germany), as in a previous study [20]. Measurements were performed at 27 °C. CDCl_3_ (St. Louis, MO, USA) was used to dissolve the *S. mukorossi* oil samples.

### 4.2. Oil Composition Analysis

This study used gas chromatography–mass spectrometry (GC-MS) to analyze the composition and content of FA found in the *S. mukorossi* seed oil. The extracted *S. mukorossi* seed oil first underwent an initial transesterification process to produce fatty acid methyl esters (FAMEs). In brief, the extracted oil was then mixed with 1 N sodium hydroxide and stirred at room temperature for 30 s. After 15 min saponification, 1 mL of boron trifluoride (in a methanol solution) (Fisher Scientific, Pittsburgh, PA, USA) was added to the sample, which was subsequently incubated at 110 °C in a dry bath for 15 min. Then, 1 mL of n-hexane was added to the sample to extract FAMEs. A GC-MS system (GCMS-QP2010, Shimadzu, Tokyo, Japan) equipped with a BPX70 capillary column (30 m × 0.25 mm i.d., 0.25 μm film thickness) was used to analyze the composition of FA found in the oil. Helium with a pressure of 75 kPa was used as a carrier gas. Samples were injected into the column with a port temperature of 250 °C. The initial oven temperature was set at 120 °C. The temperature program was carried out as follows: an initial solvent delay of 0.5 min followed by a gradual increase at a rate of 10 °C/min until samples reached 180 °C, after which the rate was reduced to 3 °C/min up to 220 °C. Finally, the temperature rate was raised to an increase of 30 °C/min until samples reached 260 °C. For mass spectrum analysis, the ionization chamber temperature was maintained at 200 °C. Compounds separated via GC were identified using MS with an electron impact (EI) mode at 70 eV. Peaks were identified through comparison against FA standards and the MS database (NIST/EPA/NIH). Percentages of fatty acid esters were derived by calculating peak area ratios.

### 4.3. Assessment of Cell Viability

In this study, Wharton-jelly-derived mesenchymal stem cells (WJMSCs), procured from BCRC (Catalog No. RM60596; Hsinchu, Taiwan), were used for the in vitro experiment. Cells were cultured in α-minimal essential medium (α-MEM; Gibco/Invitrogen, Carlsbad, CA, USA) supplemented with 20% fetal bovine serum (Gibco/Invitrogen), 1% penicillin-streptomycin (Gibco/Invitrogen), 1% nonessential amino acids (Gibco/Invitrogen), 1% sodium pyruvate (Corning, Manassas, VA, USA), 1% l-glutamine (Gibco/Invitrogen), and sodium bicarbonate. The culture was maintained at 37 °C in a CO_2_ incubator with a 5% CO_2_ atmosphere. The culture medium was refreshed every 2–3 days, and cells were subcultured upon reaching 70% to 80% confluence. Cells from passages 4 to 10 were utilized in the study.

As seed oil is hydrophobic, dimethyl sulfoxide (DMSO, Sigma-Aldrich, St. Louis, MO, USA) was employed as an emulsifier (oil/DMSO = 2:5 *v*/*v*) when mixing the oil into the culture medium. The effect of *S. mukorossi* seed oil on WJMSC viability was examined using a 3-(4,5-dimenthylthiazol-2-yl)-2,5-diphenyltetrasoliumbromide (MTT, Roche Applied Science, Mannheim, Germany) assay. WJMSCs with a density of 2 × 10^4^ cells/mL were cultured in a medium containing *S. mukorossi* seed-oil-DMSO solution. The additional concentration was 0.2% (*v*/*v*) for all the cellular experiment. At intervals of 1, 3, and 5 days of culture, a solution containing 5 mg/mL of MTT was introduced to the culture dishes and incubated for an additional 4 h. Subsequently, DMSO was added to dissolve the formazan crystals. Cell viability was assessed by measuring the optical density at a wavelength of 570 nm with a reference wavelength of 690 nm using a microplate reader (EZ Read 2000, Biochrom Ltd., Cambridge, UK).

### 4.4. The Effect of S. mukorossi Seed Oil on Calcium Deposition

WJMSCs were plated in a 24-well culture plate at a density of 2 × 10^4^ cells/mL. Once cells reached 80% confluence, the regular culture medium was substituted with an osteogenesis induction medium consisting of α-minimal essential medium (α-MEM; Gibco/Invitrogen) and 90 mM KH2PO4 (J.T. Baker, Phillipsburg, NJ, USA) supplemented with 0.1 μM Dexamethasone (Sigma-Aldrich). The medium was replaced every 2 to 3 days. The effect of *S. mukorossi* seed oil on WJMSCs osteogenesis was assessed by co-culturing the seed oil with osteogenesis medium after a 14-day culture period. Subsequently, the WJMSCs were fixed using 4% paraformaldehyde and stained with 250 µL of 2% Alizarin Red S staining solution (Alizarin Red S, Sigma-Aldrich) for 2 min. Following the removal of the staining solution, the WJMSCs were rinsed with PBS and examined using a microscope (Eclipse TS100, Nikon Corporation, Tokyo, Japan). For quantification, stained cells were treated with 10% acetic acid and 10% ammonium hydroxide and the optical density was measured spectrophotometrically at 405 nm.

### 4.5. Effect of S. mukorossi Seed Oil on Osteogenesis Gene Expression

To investigate the impact of *S. mukorossi* seed oil on osteogenesis-related gene expression in WJMSCs, the levels of alkaline phosphatase (ALP) and osteocalcin (OCN) expression were assessed using a quantitative, real-time polymerase chain reaction (qPCR) following previously described protocols [23,45]. A total of 4 μg extracted RNA was used to synthesize cDNA. The procedure was completed using a High-Capacity Reverse Transcription Kit (Applied Biosystems, Foster City, CA, USA) following the manufacturer’s instructions. The primers used for the qRT-PCR were: OCN (forward: ATGAGAGCCCTCACACTCCT; reverse: CTTGGACACAAAGGCTGCAC), ALP (forward: TTTATAAGGCGGCGGGGGT; reverse: CTGCTTTATCCCTGGAGCCC), and β-actin (forward: GAGCACAGAGCCTCGCCTTT; reverse: AGAGGCGTACAGGGATAGCA). Target cDNA was amplified utilizing a real-time DNA thermal analyzer (Rotor-gene 6000; Corbett Life Science, Sydney, Australia), employing FastStart Universal SYBR Green Master (Roche Applied Science, Mannheim, Germany). To normalize the selected fluorescence signals, the human β-actin gene was utilized as an internal control. The ΔCT of oil-treated cells was subtracted from the measured ΔCT of the control cells to obtain the ΔCT change (ΔΔCT) and fold change (2^−ΔΔCT^) of the treated cells.

### 4.6. Animal Experiment

To perform animal experiments, 200 kDa hyaluronic acid (HA) (Cheng-Yi Chemical Industry Co., Ltd., Taipei, Taiwan) at a concentration of 30 mg/mL was mixed with 10 mg/mL carboxymethyl cellulose (CMC) at a 1:1 ratio to form HA-CMC hydrogel (HG), serving as a carrier for releasing *S. mukorossi* seed oil [49]. Three male New Zealand white rabbits weighing between 3.0 and 3.5 kg were utilized as experimental subjects to test the osteogenic effect of *S. mukorossi* seed oil in vivo. These rabbits were provided with solid food and water and housed in a hygienic and controlled environment with a consistent temperature of 25 °C and humidity of 50%. Animal handling procedures and experimental protocols were formulated in accordance with guidelines provided by the National Research Council’s *Guide for the Care and Use of Laboratory Animals*. Approval for the entire protocol was granted by the Institutional Animal Care and Use Committee of the National Defense Medical Center, Taipei, Taiwan (IACUC-18-245). Animals were subjected to standard sterilization procedures before experiments were conducted. Anesthesia was achieved by intramuscular injection of Zoletil 50 (Virbac, Carros Cedex, France) at a dose of 15 mg/kg. After rabbits reached deep anesthesia, their foreheads were shaved and disinfected with povidone-iodine (Sigma-Aldrich, St. Louis, MO, USA) (Figure 9a). Four circular bone defects were prepared (6 mm in diameter) at the parietal bone (Figure 9b), as per a previous study [50,51]. In accordance with the 3R principles (Replacement, Reduction, and Refinement) outlined in the *Declaration of Helsinki*, and to mitigate experimental errors arising from inter-individual variability, twelve defects were randomly allocated between three rabbits from the HG, HG-*S. mukorossi* seed oil (HG-SM oil), and blank groups (n = 4 each). For each artificial defect, 1.0 g of prepared filling material was grafted. In order to minimize the risk of infection, rabbits were administered postoperative antibiotics and analgesics via intramuscular injection for a duration of 3 days. Following a healing period of 4 weeks, the rabbits were humanely euthanized under anesthesia (using 50 mg/mL Zoletil at a dosage of 15 mg/kg) via CO_2_ gas asphyxiation. Bone tissues at surgical sites were then cut and collected in 10% formaldehyde solution to fix and preserve the tissue. The experimental report followed ARRIVE (Animal Research: Report of in vivo Experiments) guidelines.

### 4.7. Quantification of Bone Growth In Vivo

The bone tissues in the above animal study were first checked using Micro-CT (SkyScan 1076, Bruker, Kontich, Belgium). Scanning parameters for energy level, current, and pixel size were set at 5 kV, 200 μA, and 18 μA, respectively. After reconstructing 3D images around the defects, new bone formation was calculated using analysis software (CTAn, v.1.18, Bruker). New bone was calculated as the ratio of newly formed bone volume (BV) to total defect volume (TV). Quantification of trabecular thickness, trabecular number, and trabecular separation in the defect areas were determined automatically by the software.

To obtain histological images, a decalcification procedure was performed on the collected bone tissue. Samples were decalcified using Plank–Rychlo’s solution (MUTO Pure Chemicals, Tokyo, Japan) for 7 days and then dehydrated in alcohol with gradient concentrations [51]. After embedding the samples in paraffin, sections with a thickness of 5 μm were prepared. All sample sections were stained with hematoxylin and eosin (Sigma-Aldrich, St. Louis, MO, USA) and scanned with a digital slide scanner (OPTIKA, Ponteranica, Italia). Tissue quantification was achieved using one layer of panoramic images for each sample. Percentages of new bone, connective tissue, and remaining grafting materials were measured using ImageJ software (v.1.51, National Institutes of Health, Bethesda, MD, USA).

### 4.8. Statistical Analysis

Four and six replicate samples were tested for in vitro cellular experiments and animal studies, respectively. Data are expressed as mean values ± standard deviations. Variations between samples were examined utilizing a one-way analysis of variance, and a Scheffe post hoc test was performed to demonstrate the difference. Statistical significance was defined as *p* values under 0.05 for all analyses.

## 5. Conclusions

In conclusion, this study has shown that *S. mukorossi* seed oil enhances WJMSC osteogenic differentiation and promotes bone healing. The enhanced potential for stem cell differentiation triggered by *S. mukorossi* seed oil may due to the abundance of ω-9 MUFA contained in this plant oil. Based on these results, it is reasonable to suggest that *S. mukorossi* seed oil could serve as an alternative agent for applications in bone tissue engineering.

## Figures and Tables

**Figure 1 ijms-25-06749-f001:**
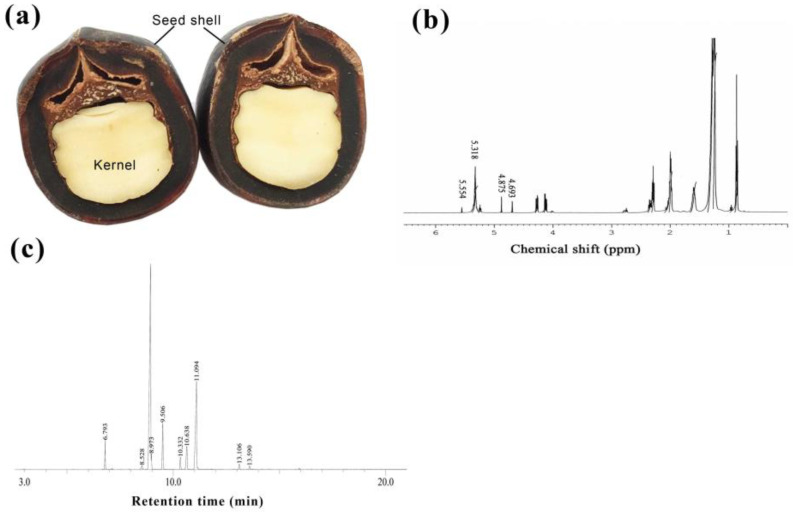
(**a**) Sagittal section of the *S. mukorossi* seed. Seed kernel is covered by a hard, black shell. (**b**) The plant species was confirmed by 1H nuclear magnetic resonance (NMR). (**c**) The composition and content of FA found in the *S. mukorossi* seed oil was detected using gas chromatography–mass spectrometry.

**Figure 2 ijms-25-06749-f002:**
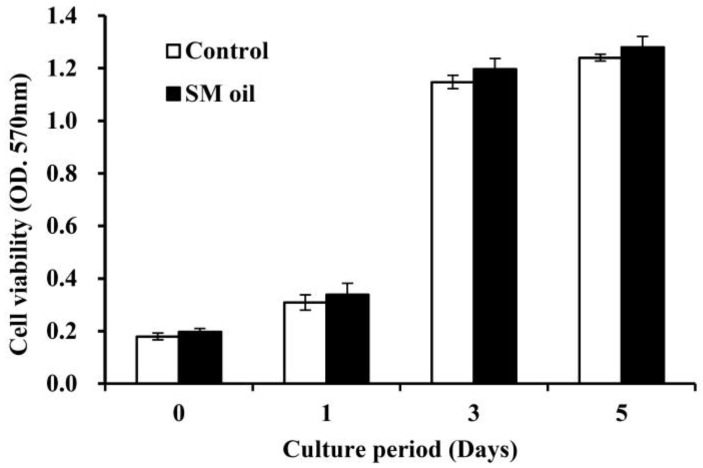
Viabilities of MSCs cultured with *S. mukorossi* seed oil for 5 days. During the incubation period, *S. mukorossi* seed oil showed no significant effect on WJMSC viability. SM denotes *S. mukorossi*.

**Figure 3 ijms-25-06749-f003:**
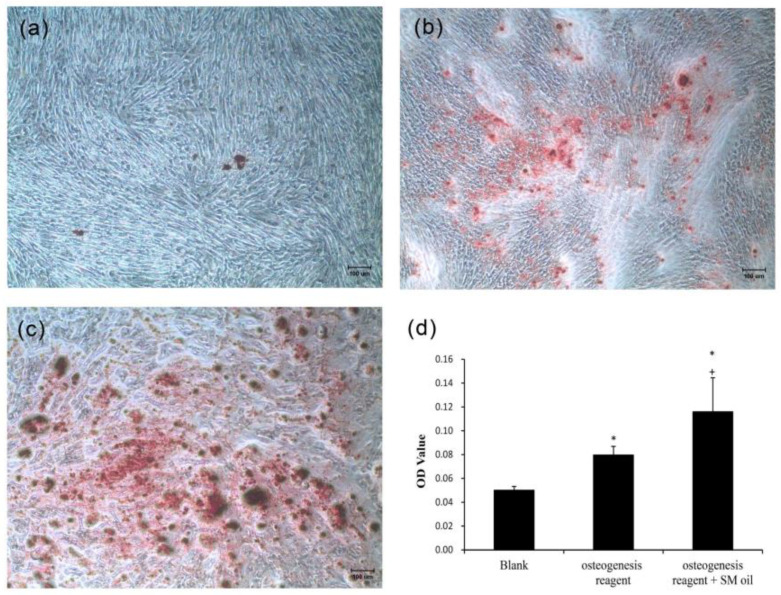
Alizarin Red S staining demonstrated that osteogenic differentiation of WJMSCs was promoted by adding *S. mukorossi* seed oil. (**a**) Mineralized nodule deposition was not observed when MSCs were cultured in standard medium. (**b**) After induction with osteogenic medium, visible deposition of mineralized nodules occurred (red spots). (**c**) After co-culturing with both osteogenic medium and *S. mukorossi* seed oil, WJMSCs secreted a greater number of mineralized nodules. (**d**) Quantification of Alizarin Red S staining reveals a notable increase in mineralized nodules due to the inclusion of *S. mukorossi* seed oil. (* and + denote comparisons with the blank group and osteogenic medium, respectively. *p* < 0.05, scale bars denote 100 μm. SM denotes *S. mukorossi*).

**Figure 4 ijms-25-06749-f004:**
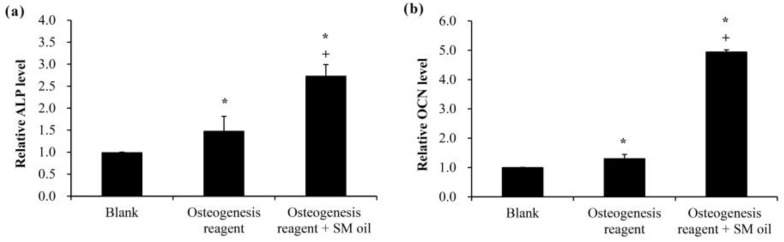
Osteogenic gene expression levels, including (**a**) ALP and (**b**) OCN, assessed following culturing of WJMSCs in standard medium, osteogenic medium, and osteogenic medium with seed oil. The addition of *S. mukorossi* seed oil significantly increased the expression level of both ALP and OCN. (* and + denote comparisons with the blank group and osteogenic medium, respectively. *p* < 0.05. SM denotes hydrogel and *S. mukorossi*).

**Figure 5 ijms-25-06749-f005:**
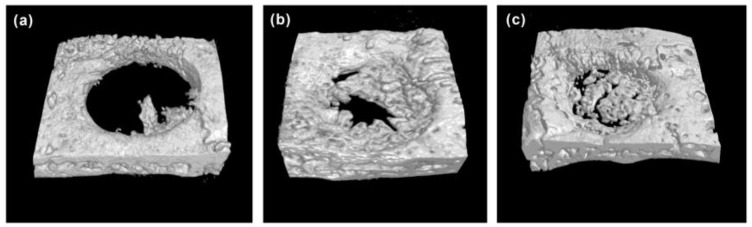
Micro-CT 3D images of bone defects grafted with various materials. Compared to the control group (**a**), both the HG and HG-SM oil group increased bone growth (**b**). The addition of *S. mukorossi* seed oil to HG increased bone growth efficiency after 4 weeks of healing (**c**).

**Figure 6 ijms-25-06749-f006:**
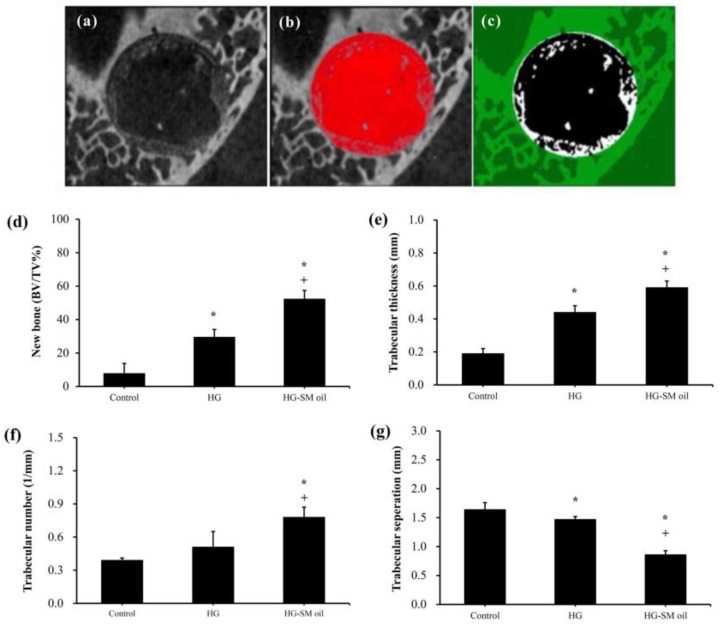
Quantification of micro-CT images (**a**–**c**) showed that adding *S. mukorossi* seed oil to bone grafts significantly increased the (**d**) efficiency of new bone formation, (**e**) trabecular thickness, (**f**) trabecular number, and (**g**) amount of trabecular separation. (* and + denote comparisons with the control and HG groups, respectively. *p* < 0.05). HG and SM denote hydrogel and *S. mukorossi*, respectively. (n = 4).

**Figure 7 ijms-25-06749-f007:**
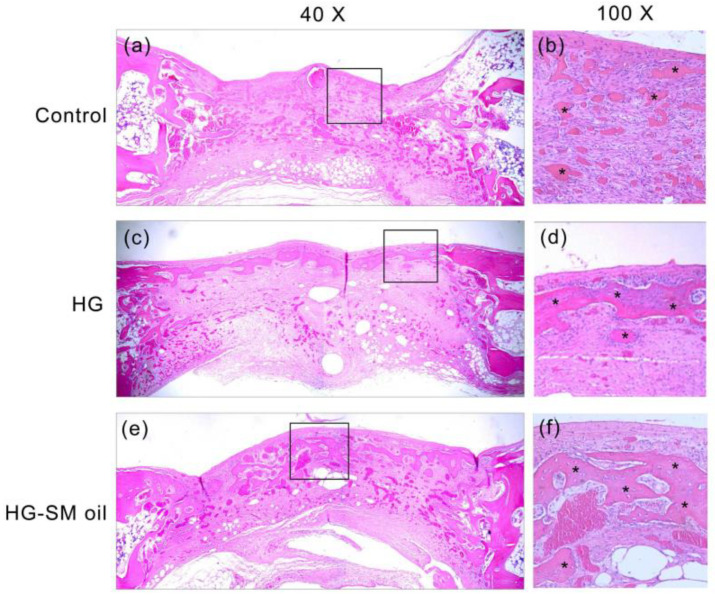
Histological images of bone tissue at artificial defects created in this study. Microscopy shows tissue grafted with blank (**a**,**b**), HG (**c**,**d**), and HG-SM oil (**e**,**f**) at 4 weeks post healing (* denotes newly formed bone). HG and SM denote hydrogel and *S. mukorossi*, respectively.

**Figure 8 ijms-25-06749-f008:**
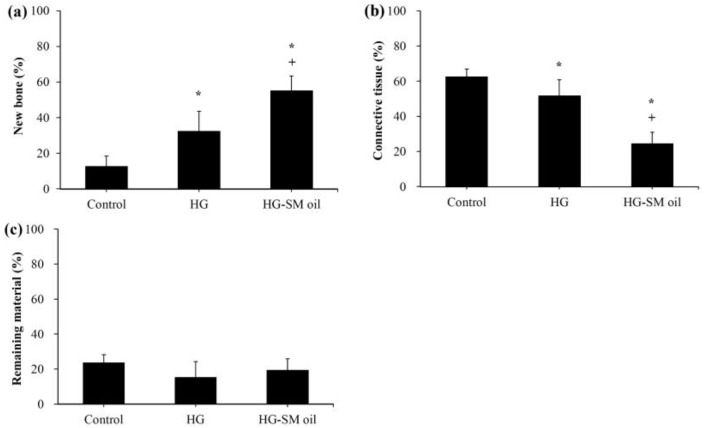
Quantitative analysis of measurements from histological images of (**a**) new bone formation, (**b**) connective tissue, and (**c**) remaining material. (* and + denote comparisons with the control and HG groups, respectively. *p* < 0.05. HG and SM denote hydrogel and *S. mukorossi*, respectively). (n = 4).

**Figure 9 ijms-25-06749-f009:**
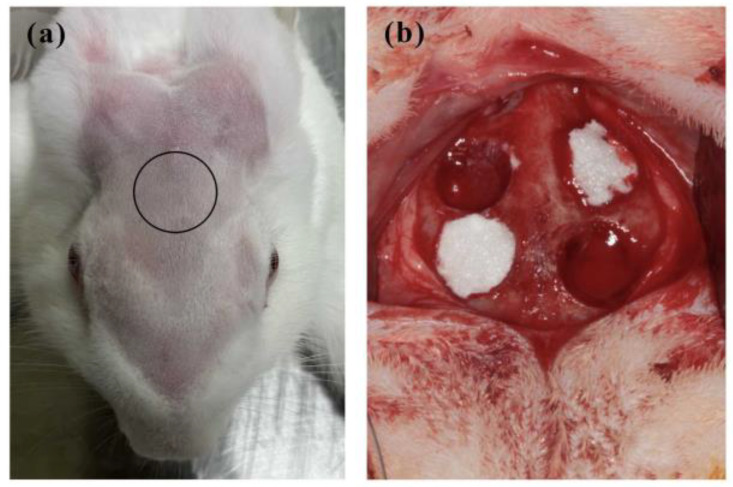
Animal experiment performed in this study. (**a**) Black circle indicates the surgical site. (**b**) Four 6 mm circular bone defects were prepared on each rabbit skull and filled with tested materials.

**Table 1 ijms-25-06749-t001:** Fatty acid composition of *Sapindus mukorossi* seed oil. SFA: saturated fatty acid. MUFA: monounsaturated fatty acid. PUFA: polyunsaturated fatty acid.

Retention Time (min)	Amount (%)	Fatty Acid	Type
6.793	3.25	Palmitic acid (16:0)	SFA
8.528	0.93	Stearic acid (18:0)	SFA
8.973	55.20	Oleic acid (18:1)	ω-9 MUFA
9.506	7.52	Linoleic acid (18:2)	ω-6 PUFA
10.332	2.04	Linolenic acid (18:3)	ω-3 PUFA
10.638	5.24	Arachidic acid (20:0)	SFA
11.094	24.08	Eicosenic acid (20:1)	ω-9 MUFA
13.106	1.03	Behenic acid (22:0)	SFA
13.590	0.7	Erucic acid (22:1)	ω-9 MUFA

## Data Availability

Data is contained within the article.

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
