# Peer review of "Effects of Sapindus mukorossi Seed Oil on Bone Healing Efficiency: An Animal Study"

_ijms, 2024, doi:10.3390/ijms25126749_

Round 1
Reviewer 1 Report
Comments and Suggestions for Authors
A carefully designed and well conducted experiment investigating the role of oily extract in osteogenesis and possible bone regeneration.
Several comments or rather suggestions are inserted below
Abstract
Very efficient in summarizing the presented work. Maybe adding two words regarding the type of animal model used -species and bone defect , location, size, would be beneficial
Introduction
Is brief to the point clearly presenting the scientific rationale for conducting a study regarding the osteogenic role of oily extracts.
Material and methods
Methods are clearly described with sufficient details. One fact that maybe is less clear, how many times /doses of oil extract were added during time in culture and especially during osteogenesis assays. Given the fact that calvaria bone in rabbits is rather thin, was not the decalcification procedure too lengthy?. Maybe choosing an additional histology staining (such as Trichrome) (or performing IHC) would better have served to document bone formation.
Discussions
Not surprisingly, WJSC did not too much proliferate (fact reported already for other cell types and different oil extracts) This might not be an issue as long as sufficient number of cells can be provided at bone defect site. The calvarian defect model has the advantage of benefiting from a rather rich vascularization as well as of lack of mechanical stress.
Do the authors consider their results could be reproduced when this two crucial factors are not fulfilled (meaning when bone defect is rather scarce vascularized and has to endure mechanical stress?
Do the authors plan to manufacture hydrogel like scaffold enhanced with oily extract? What would be the scenario of using this mixture for bone healing purposes?
To note, we do not have too much trouble with bone healing unless the defect is large, stress high or consistent pieces of bone are missing. Some sort of mechanical support should maybe envisaged.
Definitely further studies will be needed to reveal the underlying mechanism (be it calcium ions transport or improving the biophysical properties of cell membrane in terms of fluidity and stabilization will need further investigation to test such interesting hypothesis.
Author Response
Reviewer 1
- A carefully designed and well conducted experiment investigating the role of oily extract in osteogenesis and possible bone regeneration. Several comments or rather suggestions are inserted below
Author response: We sincerely thank the reviewer for his comments.
- Abstract: Very efficient in summarizing the presented work. Maybe adding two words regarding the type of animal model used -species and bone defect , location, size, would be beneficial
Author response: We sincerely thank the comment from the reviewer. The species of animal model as well as the details of bone defect were added in the abstract of the revised manuscript.
- Introduction: Is brief to the point clearly presenting the scientific rationale for conducting a study regarding the osteogenic role of oily extracts.
Author response: We sincerely thank the comment from the reviewer. On page 2, second sentence, a new reference [4] was added to the fourth paragraph to present the scientific rationale regarding the osteogenic role of oily extracts.
- Methods are clearly described with sufficient details. One fact that maybe is less clear, how many times /doses of oil extract were added during time in culture and especially during osteogenesis assays. Given the fact that calvaria bone in rabbits is rather thin, was not the decalcification procedure too lengthy? Maybe choosing an additional histology staining (such as Trichrome) (or performing IHC) would better have served to document bone formation.
Author response: We sincerely thank the comment from the reviewer. The concentration of oil-DMSO solution added to culture cells is 0.2 % (v/v) for all the cellular experiment for all the cellular experiments. These statements were added to page 10, the second paragraph of section 4.3. The time period of decalcification in this study is 7 days. We thank the reviewer for pointing out this typo. The decalcification procedure adopted in this study was designed according to a previous reference [51]. We revised the description on page 12, section 4.7.
- Not surprisingly, WJSC did not too much proliferate (fact reported already for other cell types and different oil extracts) This might not be an issue as long as sufficient number of cells can be provided at bone defect site. The calvarian defect model has the advantage of benefiting from a rather rich vascularization as well as of lack of mechanical stress. Do the authors consider their results could be reproduced when this two crucial factors are not fulfilled (meaning when bone defect is rather scarce vascularized and has to endure mechanical stress?
Author response: We sincerely thank the reviewer for his comments. Mechanical stress-induced bone regeneration is indeed linked to the deformation of the membrane, which alters its fluidity. Similarly, the ontogenesis effect of the oil also involves changes in membrane fluidity. Therefore, the bone regeneration effect of the oil can be complementary in situations where stress is lacking. We have added a paragraph discussing this issue on page 9, the third paragraph. However, regarding the relationship between plant oil and vascularization, we could not find sufficient references to support this.
- Do the authors plan to manufacture hydrogel like scaffold enhanced with oily extract? What would be the scenario of using this mixture for bone healing purposes? To note, we do not have too much trouble with bone healing unless the defect is large, stress high or consistent pieces of bone are missing. Some sort of mechanical support should maybe envisaged.
Author response: We sincerely thank the reviewer for his suggestion. The limitation of application of oil for bone healing is its high fluidity that results in flow out of the defect which may reduce its bone growth efficiency. Thus manufacture a hydrogel scaffold to release this oil is a necessary protocol for applying this material in clinical. In the revised manuscript, we added this descript on page 9, the first paragraph.
- Definitely further studies will be needed to reveal the underlying mechanism (be it calcium ions transport or improving the biophysical properties of cell membrane in terms of fluidity and stabilization will need further investigation to test such interesting hypothesis.
Author response: We sincerely thank the reviewer for his suggestion. In the revised manuscript, the possible further studies of application this oil for bone healing was added on page 9, the last sentence of third paragraph.
Reviewer 2 Report
Comments and Suggestions for Authors
I read with interest the paper entitled “Effects of Sapindus mukorossi seed oil on bone healing efficiency: an animal study”.
The authors performed and reported data regarding the extraction of Sapindus mukorossi seed oils, their chemical composition analysis, preliminary in vitro investigations on Warton’s Jelly derived mesenchymal stem cells and in vivo bone healing capacity in New Zealand white rabbits.
The topic is interesting, original and novel, since very few papers are available in literature.
I have several concerns aimed at improving the manuscript, as following.
My major issue is related to the performed statistical analysis, since the authors used the Student’ t test even if both in in vitro and in vivo tests there were more than two groups to be compared or different experimental times. Thus, the Student’s t test is unsuitable; moreover, the choice of a parametric or non-parametric test is usually done after the analysis of variance.
Without a suitable statistical analysis, it’s impossible to evaluate and interpret correctly results.
According to the ARRIVE guideline, there is the lack of the statistical test and method used to determine the animal sample size. Used animals were male or female? How many replicates have been performed in in vitro assays? On how many sections histomorphometric measurements have been performed? These methods should be added to give scientific soundness and robustness to the paper.
The second concern, that need to be further and better reported, is about the selected concentration of MS oil. How was the dose selected for both in vitro and in vivo tests? How was the MS oil sterilized? Why in the in vivo study MS oil was added to a 1:1 HA-CMC gel? The cytotoxicity and biocompatibility of HA-CMC gel have been previously established?
Thirdly, the authors reported the fatty acid composition contained in Sapindus mukorossi seed oil, founding substances (SFA) able to stimulate osteoclastogenesis and other (w-6 PUFA) able to stimulate inflammation. I think that the investigation of these parameters both in vitro and in vivo tests (for example, by osteoclastogenesis and inflammatory related genes analysis and/or immunohistochemical staining on bone sections) should be added to claim “…reduces the negative effects of using S mukorossi seed oil as material for bone regeneration (lines 223-224)”.
Minor suggestions
The sentence at lines 45-46 needs reference.
The sentence at lines 55-56 is repeated at lines 58-59
line 121 it seems that some lines are missing
line 428: how was the residual material identified?
line 437 the author wrote “w-6 UFA” instead of w-9
Legend for Figure 6. There is inconsistency between the letters composing the panel (a-g) and the legend (a-d)
Lines 165-167: The authors claim that new bone (Fig 6d) accounts for 3.79 and 6.72 in defects treated with HG and HG-SM oil, respectively; however, in Fig 6 d BV/TV values are about 30 and 50% for HG and HG-SM. Please, check it carefully.
lines 169-170: “No disparity in trabecular number... or separation…was noted with the use of HG alone” but in Graph 6f and 6g, significant differences are depicted. This should be checked and verified by performing the correct statistical analysis, as suggested.
Comments on the Quality of English Languagenone
Author Response
Reviewer 2
- I read with interest the paper entitled “Effects of Sapindus mukorossi seed oil on bone healing efficiency: an animal study”. The authors performed and reported data regarding the extraction of Sapindus mukorossi seed oils, their chemical composition analysis, preliminary in vitro investigations on Warton’s Jelly derived mesenchymal stem cells and in vivo bone healing capacity in New Zealand white rabbits. The topic is interesting, original and novel, since very few papers are available in literature. I have several concerns aimed at improving the manuscript, as following.
Author response: We sincerely thank the reviewer for his comments.
- My major issue is related to the performed statistical analysis, since the authors used the Student’ t test even if both in in vitro and in vivo tests there were more than two groups to be compared or different experimental times. Thus, the Student’s t test is unsuitable; moreover, the choice of a parametric or non-parametric test is usually done after the analysis of variance. Without a suitable statistical analysis, it’s impossible to evaluate and interpret correctly results.
Author response: We sincerely thank the reviewer for his comments. The statistical analysis was reperformed by using ANOVA analysis in the revised stage. Based on the results, the major results and conclusions have not changed. In the revised manuscript, we rewrite the statistical analysis method on page 12, section 4.8.
- According to the ARRIVE guideline, there is the lack of the statistical test and method used to determine the animal sample size. Used animals were male or female? How many replicates have been performed in in vitro assays? On how many sections histomorphometric measurements have been performed? These methods should be added to give scientific soundness and robustness to the paper.
Author response: We sincerely thank the reviewer for his comments. To follow the 3R spirit of the Declaration of Helsinki and to eliminate artifacts due to experimental error caused by inter-individual differences, twelve defects on three male rabbits were randomly assigned to prepared HA, HA/oil, and control groups (n = 4). This information was added on page 11, section 4.6. The sample size for in vitro cellular experiments and animal study are six and four, respectively. This information was added to section 4.8 on page 12. In animal study, 1.0 g of prepared filling material was grafted to the artificial defects. This information was added to section 4.6 on page 11. For histomorphometric measurements, bone quantification was achieved using one layer of panoramic images for each sample. This information was added to section 4.7 on page 12.
- The second concern, that need to be further and better reported, is about the selected concentration of MS oil. How was the dose selected for both in vitro and in vivo tests? How was the MS oil sterilized? Why in the in vivo study MS oil was added to a 1:1 HA-CMC gel? The cytotoxicity and biocompatibility of HA-CMC gel have been previously established?
Author response: We sincerely thank the reviewer for his comments. The concentration of SM oil used in cellular experiments is 0.2 % (v/v). This information was added to page 10, the second paragraph of section 4.3. In animal study, 1.0 g of prepared filling material was grafted to the artificial defects. This information was added to section 4.6 on page 11.
To avoid damaging the contents of the oil and because the oil exhibits excellent anti-bacterial properties (Int. J. Mol. Sci. 2019, 20, 2579), the SM oil was not treated with a sterilizing procedure alone before the experiment. However after it was added to medium, it received standard sterilizing procedure of medium preparation.
The limitation of application of oil for bone healing is its high fluidity that results in flow out of the defect which may reduce its bone growth efficiency. Thus in this study we manufacture a hydrogel scaffold to release this oil. We added this limitation on page 8, fourth paragraph.
The cytotoxicity and biocompatibility of HA-CMC gel used in this study have been established by a previously paper (Biomed Res Int. 2016; 2016: 3640182). We added this paper as a reference [49] in the revised manuscript.
- Thirdly, the authors reported the fatty acid composition contained in Sapindus mukorossi seed oil, founding substances (SFA) able to stimulate osteoclastogenesis and other (w-6 PUFA) able to stimulate inflammation. I think that the investigation of these parameters both in vitro and in vivo tests (for example, by osteoclastogenesis and inflammatory related genes analysis and/or immunohistochemical staining on bone sections) should be added to claim “…reduces the negative effects of using S mukorossi seed oil as material for bone regeneration (lines 223-224)”.
Author response: We sincerely thank the reviewer for his comments. We did find SM oil contains SFA and w-6 PUFA. However, we did not test whether these FAs really can induce osteoclastogenesis and inflammation. The possible negative functions of these materials are according to previous papers. Since the purpose of this study is not to claim the relationship between SFA/w-6 PUFA and osteoclastogenesis /inflammation, we revised our statement to “…which avoid the possible side effects of using S. mukorossi seed oil as a material for bone regeneration.” in the revised manuscript (on page 8, first paragraph).
Minor suggestions
Author response: We sincerely thank the reviewer for all his comments.
- The sentence at lines 45-46 needs reference.
Author response: On page 2, second line, a new reference [4] (Materials 2021, 14, 1867) was added in the revised manuscript.
- The sentence at lines 55-56 is repeated at lines 58-59
Author response: On page 2, second paragraph, the sentence was revised to ” ….the effects of ω-3 and ω-6 UFA on bone health are different.”.
- line 121 it seems that some lines are missing
Author response: On page 4, second patagraph, the intact sentence should be ” …. to exert a significant effect on the viability of WJMSC cells.” In the revised manuscript, we added the space between figure and text to avid this mistake.
- line 428: how was the residual material identified?
Author response: The remaining grafting material was auto-detected by the software according to gray level of the image.
- line 437 the author wrote “w-6 UFA” instead of w-9
Author response: We sincerely thank the reviewer for pointing out this typo error. On page 12, section 5, we corrected the word “w-6” to “w-9” in the revised manuscript.
- Legend for Figure 6. There is inconsistency between the letters composing the panel (a-g) and the legend (a-d)
Author response: We sincerely thank the reviewer for pointing out this typo error. On page 6, the legend for Figure 6, the mismatch was revised.
- Lines 165-167: The authors claim that new bone (Fig 6d) accounts for 3.79 and 6.72 in defects treated with HG and HG-SM oil, respectively; however, in Fig 6 d BV/TV values are about 30 and 50% for HG and HG-SM. Please, check it carefully.
Author response: In Figure 6, the new bone formation was expressed as BV/TV %. The numbers 3.79 and 6.72 in defects refer to the multiples of the new bone values ​​of these two groups compared to the control.
- lines 169-170: “No disparity in trabecular number... or separation…was noted with the use of HG alone” but in Graph 6f and 6g, significant differences are depicted. This should be checked and verified by performing the correct statistical analysis, as suggested.
Author response: We sincerely thank the reviewer for pointing out this typo error. The statistical analysis was recalculated using ANOVA. The results show no change. On page 6, the first sentence, the mismatch between the figure and text was revised.
Reviewer 3 Report
Comments and Suggestions for Authors
The authors’ group has previously published osteogenic effect of S. mukorossi seed oil on dental pulp stem cells. In this study, they investigated its osteogenic effect on human umbilical cord Wharton jelly stem cells. Additionally, they evaluated its effect on bone healing using a rabbit artificial bone defect model. Their experimental flow was straightforward, and the results revealed drastic effects of S. mukorossi seed oil. The authors discussed the potential mechanisms underlying these effects well. I have a few comments and requests that require the authors’ attention and revision.
1. To better understand the effect of S.mukorossi on osteogenesis , it would be worthwhile to examine the expression profiles of the following osteogenic genes: Runx2, Osteopontin, Osterix, Col1a1, and bone sialoprotein.
2. English editing, specifically in grammar and style, is required to improve readability.
Comments on the Quality of English LanguageModerate editing of English language required.
Author Response
Reviewer 3.
The authors’ group has previously published osteogenic effect of S. mukorossi seed oil on dental pulp stem cells. In this study, they investigated its osteogenic effect on human umbilical cord Wharton jelly stem cells. Additionally, they evaluated its effect on bone healing using a rabbit artificial bone defect model. Their experimental flow was straightforward, and the results revealed drastic effects of S. mukorossi seed oil. The authors discussed the potential mechanisms underlying these effects well. I have a few comments and requests that require the authors’ attention and revision.
Author response: We sincerely thank the reviewer for his comments.
- To better understand the effect of S.mukorossi on osteogenesis , it would be worthwhile to examine the expression profiles of the following osteogenic genes: Runx2, Osteopontin, Osterix, Col1a1, and bone sialoprotein.
Author response: We sincerely thank the reviewer for his comments. The gene expression analysis is important for clarifying the mechanism of this study. However, we cannot achieve this goal at this stage because the samples were not collected. We added this issue in limitation of this study and mentioned it to be a direction of further study (on page 9, third paragraph).
- English editing, specifically in grammar and style, is required to improve readability.
Author response: We sincerely thank the reviewer for his comments. The English writing was corrected by a by a native English teacher at the revision stage.

Round 2
Reviewer 2 Report
Comments and Suggestions for Authors
Two issues remained unanswered/critical:
tha authors wrote numers of replicates performed in vitro and in vivo. However, they didn't perform an a priori statistical analysis to determine the sample size. This is a limitation of the study.
Secondly, if I understood, histomorphometric measurements have been performed only on one section for each sample. Usually histomorphometric measurements are performed on at least three sections representative of each sample, to avoid bias related to the analysed ROI. This aspect might affect results and represents a limitation of the study
Author Response
Reviewer 2.
- tha authors wrote numers of replicates performed in vitro and in vivo. However, they didn't perform an a priori statistical analysis to determine the sample size. This is a limitation of the study.
Author response: We sincerely thank the reviewer for his comments. We did not perform a priori statistical analysis to determine the sample size. The number of replicates performed in vitro and in vivo experiments was according to previous studies. We agree an advanced experiment with a large sample size is needed for future study. This limitation was added to the last sentence of Discussion on page 9.
- if I understood, histomorphometric measurements have been performed only on one section for each sample. Usually histomorphometric measurements are performed on at least three sections representative of each sample, to avoid bias related to the analysed ROI. This aspect might affect results and represents a limitation of the study
Author response: Given the fact that calvaria bone in rabbits is rather thin, it is difficult to make three sections. We agree this may result in difficult for evaluating the bone growth situation at deep tissue. We added this as a limitation to the last sentence of page 9, paragraph 1.
